# Heparan Sulfate Proteoglycans in Tauopathy

**DOI:** 10.3390/biom12121792

**Published:** 2022-11-30

**Authors:** Yanan Zhu, Lauren Gandy, Fuming Zhang, Jian Liu, Chunyu Wang, Laura J. Blair, Robert J. Linhardt, Lianchun Wang

**Affiliations:** 1Department of Molecular Pharmacology & Physiology, Byrd Alzheimer’s Research Institute, Morsani College of Medicine, University of South Florida, Tampa, FL 33612, USA; 2Center for Biotechnology and Interdisciplinary Studies, Department of Chemistry and Chemical Biology, Department of Biological Sciences, Rensselaer Polytechnic Institute, Troy, NY 12180, USA; 3Eshelman School of Pharmacy, The University of North Carolina at Chapel Hill, Chapel Hill, NC 27599, USA; 4Department of Molecular Medicine, Byrd Alzheimer’s Research Institute, Morsani College of Medicine, University of South Florida, Tampa, FL 33613, USA

**Keywords:** tauopathy, heparan sulfate proteoglycan, tau, Alzheimer’s disease

## Abstract

Tauopathies are a class of neurodegenerative diseases, including Alzheimer’s disease, and are characterized by intraneuronal tau inclusion in the brain and the patient’s cognitive decline with obscure pathogenesis. Heparan sulfate proteoglycans, a major type of extracellular matrix, have been believed to involve in tauopathies. The heparan sulfate proteoglycans co-deposit with tau in Alzheimer’s patient brain, directly bind to tau and modulate tau secretion, internalization, and aggregation. This review summarizes the current understanding of the functions and the modulated molecular pathways of heparan sulfate proteoglycans in tauopathies, as well as the implication of dysregulated heparan sulfate proteoglycan expression in tau pathology and the potential of targeting heparan sulfate proteoglycan-tau interaction as a novel therapeutic option.

## 1. Introduction

Tauopathies are a heterogenous family of progressive neurodegenerative diseases featured with the deposition of abnormally folded species of the microtubule-associated protein tau (tau) mainly in neurons, glia, and extracellular space [1]. There are 26 tauopathies identified, including Alzheimer’s disease (AD), frontotemporal dementia with parkinsonism-17 (FTDP-17), corticobasal degeneration (CBD), progressive supranuclear palsy (PSP), argyrophilic grain disease (AGD), Pick’s disease (PiD), and other diseases where tau plays a major role. Based on the major tau isoforms contained in the tau deposits, tauopathies are classified into 3 repeat (R) tauopathies, 4R tauopathies, and 3R+4R tauopathies (with approximately an equal ratio of 3R tau and 4R tau) [1,2,3]. The diseases are also classified into primary and secondary tauopathies [1,3]. The primary tauopathies are a group of neurodegenerative diseases in which tau is believed to be the major contributing factor of the neurodegenerative process, such as PiD, PSP, CBD, and AGD. The secondary tauopathies are featured with tau aggregation as a response to other pathological proteins or events, like amyloid beta (Aβ) in AD and repetitive brain injury in the chronic traumatic encephalopathy [1]. In this review, we provide a brief overview of the structure and expression of tau and its physiological and pathological functions in tauopathy and then focus on the interaction of tau with heparan sulfate proteoglycan and related pathological processes.

## 2. The Tau Protein

Tau protein belongs to the microtubule-associated proteins family [4]. Tau is found predominantly in the axon of the adult neurons and at low levels in the cell body, dendritic spines, and axonal presynaptic terminal [4]. The subcellular distribution of the tau is developmentally and environmentally regulated and isoform-dependent [5,6,7,8]. In addition, tau is detected at low levels in glial cells and outside cells [2,7,9]. The various subcellular, extracellular, and cell-type expressions indicate that tau may play various functions under different circumstances. Human tau is encoded by a single gene, microtubule-associated protein tau (*Mapt*, located on chromosome 17q21. *Mapt* gene comprises 16 exons (Figure 1. By alternative splicing of exon 2 (E2), exon 3 (E3), and exon 10 (E10), six isoforms of tau are expressed in the normal adult human brain [2]. The largest isoform contains 441 amino acid residues, including an N-terminus projection domain with two inserts (N1 and N2), a proline-rich domain, and a microtubule-binding repeat domain composed of four repeat motifs (R1–R4) that mediate microtubule-binding and tau aggregation [2]. The isoforms differ in the absence or presence of N1, N2, and R2 domains leading to the following 6 isoforms: 2N4R, 1N4R, 0N4R, 2N3R, 1N3R, and 0N3R (Figure 1) [1,3,10]. The N1 and N2 are encoded by E2 and E3, respectively, and R2 is encoded by E10. Besides the six tau isoforms, researchers also widely used another two truncated tau proteins, the K18 and K19, which contain only the four and three microtubule-binding domains, respectively (Figure 1). Given the differential distribution of tau in different cell compartments and various tau isoforms, tau likely plays different functions in different environments.

## 3. Tau in Physiological States 

As a microtubule-associated protein, Tau promotes axonal outgrowth through stabilizing neuronal microtubules [2]. Early studies indeed showed that tau stabilizes the axonal microtubules, promotes the assembly of microtubules, and regulates the dynamic instability of the microtubules (Figure 2A) [11,12,13], suggesting that tau is critical for developing a healthy neuron. However, knockout of mouse *Mapt*, the gene encoding tau protein, does not lead to a severe developmental defect or overt abnormalities at young ages [14,15,16]. Additionally, in neuronal cell line studies, tau was shown to inhibit axonal transport through multiple mechanisms, including competing with kinesin or dynein for binding to microtubules [17], competing with other cargos for binding to kinesin [18,19], reducing the number of cargo-associated kinesin motors [20], and releasing cargos from the kinesin chains [21] (Figure 2A). However, knockout or overexpression of tau does not alter axonal transport in cultured primary neurons [22,23,24]. These observations suggest that tau might be dispensable for microtubule assembly, stability, and axonal transport. The lack of the expected microtubule and axonal transport defects in the *Mapt^−/−^* mice is likely due to redundancy in function among tau and compensatory effect from other microtubule-associated proteins for the loss of tau [15,25]. Tau has been suggested to regulate synaptic physiology and plasticity, although it is expressed at a low level in the dendrites (Figure 2A) [26,27,28]. Tetsuya et al. observed a selective deficit in long-term depression (LTD) but not long-term potential (LTP) in *Mapt^−/−^* mice [27]. A study reported by Ahmed et al. observed a severe deficit in LTP, but no change in LTD, in another *Mapt^−/−^* mice [28]. Tau is also expressed in the nuclei and is thought to maintain the integrity of the genomic DNA (Figure 2A) [29]. Recent studies have emerged that tau may generally act as a scaffolding protein interacting with various kinases and phosphatases to regulate phosphorylation-based signaling pathways, functioning as a signaling hub protein within cells [30].

The *Mapt*^−/−^ mice studies have uncovered that tau functionally involves neurogenesis, locomotor function, and the learning and memory [31]. In neurogenesis studies, various lines of *Mapt*^−/−^ mice were reported with contradictory results, showing that tau deficiency either decreased [16,32] or increased [14,33,34] neurogenesis reflected by the neuroD and DCX positive cells in the mouse brain. In behavioral studies, *Mapt*^−/−^ mice displayed impairments to motor function, learning, and memory at 10–11 weeks old [35] and marked motor deficits at old age (12–20 months) [36,37]. However, other studies reported normal learning, memory, and motor function in middle-aged *Mapt*^−/−^ mice [14,38,39,40]. Although the observations are inconsistent, these studies do demonstrate that tau critically plays vital physiological functions in the CNS.

## 4. Tau in Pathological States

Physiological tau is a natively unfolded and highly soluble protein with little tendency for aggregation [9]. Under pathogenic conditions, the dynamics and equilibrium of tau-microtubule binding are disrupted, leading to tau aggregation and oligomerization into paired helical filaments (PHFs) and, further, the neurofibrillary tangles (NFTs) which accumulate in neurons, glia, and extracellular space (Figure 2B) [2]. The formation of tau aggregates is more strongly correlated with cognitive decline than the distribution of senile plaques formed by amyloid beta (Aβ) protein deposits, another pathological hallmark of AD [41]. Currently, we partially understand the underlying mechanisms of tau pathology and tau-mediated neurodegeneration with most information primarily obtained from AD studies.

### 4.1. Tau Mutations 

In primary tauopathies, sporadic cases constitute most of the incidence, with nearly 31% of the patients having a family history [1,42]. Only 5–10% of the familial inheritance is associated with *Mapt* gene mutations [42], while in secondary tauopathies, including AD, no pathogenic *Mapt* gene mutation has been found [1,42]. Currently, 112 mutations in the *Mapt* gene have been discovered, and most of the mutations occur in the microtube-binding domains [1,43] (https://www.alzforum.org/mutations/mapt (accessed on 1 July 2022)). Some of these mutations are pathogenic, causing alterations in tau isoform production and microtubule dynamics to potentiate tau aggregate formation, as seen in FTDP-17, CBD, and PSP [2].

The alternative splicing of the *Mapt* gene E10 generates 3R- or 4R-tau isoforms, which function differently in the polymerization and stabilization of neuronal microtubules [2]. Due to an extra microtubule-binding repeat, 4R-tau binds more effectively to microtubules and stimulates the assembly of microtubules [44]. The alternative splicing of E10 results in a highly self-complementary stem-loop at the intron-exon interface. This structure prevents the binding of the U1 small nuclear RNA (snRNA), resulting in the E10 inclusion and 4R tau expression [44] and maintenance of normal 3R and 4R tau ratio [44]. Under normal conditions, the E10 alternative splicing results in approximately equal levels of 3R-tau and 4R-tau in the brain. Several intronic pathogenic mutations in E10 disrupt or destabilize the highly self-complementary stem-loop to make this region more available for U1 snRNA, thereby increasing E10 inclusion and 4R-tau expression resulting in 4R tau dominant tauopathies, such as PSP and CBD [44]. In addition, some mutations in E10, such as ΔK280, P301L, V337M, and R406W, attenuate tau microtubule binding and assembly functions and increase tau’s propensity to aggregate [43]. In addition, the other mutations, such as Q336H and Q336R in E12, reduce tau phosphorylation and enhance tau binding to microtubules but still lead to an increased tau aggregation [45].

Furthermore, mutations outside the microtubule-binding domains impact tau activity. The E1 mutations R5H and R5L and the E7 mutation A152T decrease tau binding to the microtubule [43]. Therefore, the pathogenic tau mutations which affect R3-and R4-tau isoform balance and tau’s binding activity to microtubules increase PHFs and NFTs formation in tauopathy.

### 4.2. Tau Post-Translational Modifications 

Phosphorylation is a physiological post-translational modification that modulates the binding of tau to microtubules [46]. Tau within NFTs is often hyperphosphorylated, and the hyperphosphorylated tau loses its affinity for microtubules and tends to self-assemble into oligomers, PHFs, and NFTs in the cytosol, indicating that the increased phosphorylation represents one common factor that strongly correlates with tau aggregation under pathological conditions [2,47]. Tau possesses as many as 85 potential phosphorylation sites in the longest tau isoform (2N4R), and most of these sites are accessible for phosphorylation owing to its unfolded structure [2]. Tau hyperphosphorylation is strongly associated with tauopathies including AD [2,46,47].

Studies have indicated that certain residues are normally phosphorylated in a non-pathogenic state, such as Thr231, potentially by GSK3β [48]. Mapping phosphorylation sites on tau in AD patients determined that phosphorylation at several residues, such as Tyr18, Ser199, Ser202, Thr205, Thr231, and Ser422, was increased [49]. Other studies observed that phosphorylation at S214, S258, S262, S293, S305, S324, and S356 inhibits tau aggregation, while phosphorylation at T149, T153, S199, S202, T205, and T212 increases tau aggregation [50]. At present, the contribution of phosphorylation of each potential modification site in tau aggregation in vivo and neurodegeneration remains unknown.

Studies have also identified acetylation, methylation, ubiquitination, sumoylation, and glycosylation as post-translational modifications (PTMs) of tau that either enhance or inhibit tau aggregation and degradation [51,52,53,54,55,56,57]. PTMs can stabilize tau monomers or protofilaments, leading to greater proximity and aggregation. PTMs have also been shown to compete with or promote further modification. For example, sumoylation of tau by SUMO-1/E3 SUMO ligase was shown to compete for the same sites as ubiquitination, while simultaneously promoting hyperphosphorylation of tau (and vice versa) [58]. *N*-glycosylation of tau has also been implicated in facilitating phosphorylation of tau via protein kinase A (PKA), as glycosylated tau was shown to be a better substrate than non-glycosylated tau protein for PKA [59].

Furthermore, tau is subject to cleavage by various enzymes such as caspase 3, which cleaves tau behind Asp421, or asparagine endopeptidase, which cleaves tau at Asn255 and Asn368 [60,61]. Specific truncations differ between normal and AD brains and a tauopathy mouse model [60,61]. Normal brains contain region-specific C-terminal and N-terminal truncated tau [62]. Many of these truncations disrupt the structure of normal tau, increase its propensity to aggregate, and show an augmented ability to recruit tau [60,61,63,64]. Notably, certain N-terminal truncations increase tau’s ability to bind and stabilize microtubules, indicating a potential physiological function of the tau truncation [62]. Truncation has also been shown to impact site-specific phosphorylation [64] and may modulate other PTMs. These studies demonstrate that alteration of post-translational modification modulates various pathological processes in tauopathy and may represent the most acquired causation factors of the disease.

### 4.3. Tau Seed Propagation

Tauopathies show pathological hallmarks in the brain. AD is the most common tauopathy, characterized by the sequential deposition and spread of aberrant tau protein in a predictable pattern that correlates with clinical severity [2,65,66]. As described in the Braak criteria, AD progression can be classified into six stages, beginning with the appearance of initial tau lesions in the transentorhinal cortex during stage I. During the subsequent stages of disease progression, the density of tau lesions increases, and NFTs spread to the entorhinal cortex in stage II, then to limbic regions of the brain in stage III, before finally reaching the neocortex in stage IV and beyond [67,68]. The propagation of tau in the brain suggests that pathogenic tau can spread prion-likely, passing from diseased neurons to healthy neurons, which act as seeds to template misfolding and aggregation (Figure 2B) [2,69]. This has been supported by complementary in vitro cellular models and in vivo animal and patient studies [70,71,72,73,74]. For example, different tau forms released from pre-synaptic neurons can be internalized by the post-synaptic neurons through multiple molecular mechanisms, showing a prion-like disease progression in the mouse brain [75]. A recent study identified four distinct trajectories of tau deposition in Alzheimer’s disease and suggested that pathology originates and spreads through distinct corticolimbic networks in the different subtypes, implying that different molecular mechanisms might be involved [73].

### 4.4. Tau-Mediated Neurotoxicity

Animal models prove that tau defects can cause synaptic damage in mice [76,77] and *Drosophila* [78]. The transgenic PS19 mice overexpressing human tau P301S (a mutation that can cause FTDP-17) develop hippocampal synaptic loss by three months of age before NFTs formation, showing a prominent decrease in levels of the pre-synaptic proteins, synaptophysin and β-synuclein in CNS [76]. At six months old, an age that precedes marked NFTs formation and neuronal loss, PS19 mice develop impaired synaptic conduction, presynaptic function, and LTP in the CNS [76]. Similarly, other tau transgenic mouse lines have shown a reduced number of spine synapses in the absence of NFTs formation [77], and the accumulation of early-stage aggregated tau species is associated with the development of functional deficits during the tauopathy progression [79]. These observations prove that tau dysfunction induces neurotoxicity and neurodegeneration and suggest that the formation of tau oligomers, the PHFs, can lead to synaptic loss.

Currently, several mechanisms have been suggested regarding the transition from normal tau to toxic tau, including: (1) Alteration of binding affinity of tau to the microtubule. The altered microtubule-binding activity that leads to either increased or reduced tau binding essentially blocks the movement of motor protein and results in improper distribution of tau in the brain [80,81]. (2) Local tau elevation caused by mis-sorting as shown in the tau transgenic mice. The high levels of tau cause microtubules to bundle and impede mitochondrial movement, leading to mitochondrial degeneration, loss of ATP, and synaptic degeneration [81]. Additionally, high levels of unbound tau may compete with potential kinesin cargo and thus prevent their translocation to the synapse [1,19,82,83]. (3) Tau filament deposition. The formation of PHFs and deposits of NFTs in the cytosol may physically obstruct the movement of mitochondria along microtubules or inhibit fast axonal transport by triggering the release of cargo from the kinesin [84]; and (4) Dysfunctional tau increases the susceptibility of neurons to Aβ and excitotoxic insults, such as the excessive activation of glutamate receptors, supporting tau as a downstream mediator of Aβ-induced toxicity in AD [85,86]. Deciphering the causes and effects of tau-mediated toxicity appears complex, as evidenced by the tau transgenic and knockout animal studies, which have suggested diverse, and sometimes conflicting mechanisms of tau neurotoxicity [70]. Some of the inconsistencies may reflect differences among tau mutations, isoforms, abnormal modifications, the solubility of tau, tau expression levels, and intermediation of neurotoxic signals in the experimental models employed to examine tau-mediated neurodegeneration.

## 5. Heparan Sulfate Proteoglycans

Heparan sulfate proteoglycans (HSPGs) are macromolecules ubiquitously expressed in mammalian tissues. They are comprised of a core protein to which one or more HS glycosaminoglycan (GAG) chains are covalently attached [87,88]. The HSPGs are classified based on the location of their core proteins (Figure 3A). Syndecans (SDCs) and glypicans are the two major membrane-bound PGs that are linked to the plasma membrane by a transmembrane domain or a glycosylphosphatidylinositol (GPI) linker, respectively. In addition, there are three “part-time” cell-surface PGs, betaglycans, neuropilins, and CD44v3, which do not always have an HS chain moiety and are located on the cell surface through their transmembrane domains. Other HSPGs include agrin, perlecan, type XVIII collagen in the extracellular matrix (ECM), and serglycin in the intracellular secretory vesicles (Figure 3A) [87,88].

Most of the biological functions of HSPGs are mediated by their HS chains [87]. The HS chain is a linear polysaccharide containing 50–200 disaccharide repeats composed of uronic acid (either glucuronic acid (GlcA) or iduronic acid (IdoA)) and N-acetylglucosamine (GlcNAc) (Figure 3B). The biosynthesis of HS occurs in the endoplasmic reticulum and the Golgi apparatus and involves various enzymes. HS biosynthesis occurs in three major steps: chain initiation, elongation, and modification [88]. Before HS biosynthesis, the xylose (Xyl) residue of a tetrasaccharide linker, GlcA-galactose (Gal)-Gal-Xyl, is covalently linked to a selected serine residue in the core protein [89]. HS biosynthesis is initiated by exostosin-like glycosyltransferase 3 (Extl3), which attaches the first GlcNAc residue to the GlcA residue of the linker to form the first GlcNAc-GlcA disaccharide repeat, followed by Ext1/Ext2 that alternately adds GlcA and GlcNAc to extend the HS chain. Meanwhile, the nascent HS chain undergoes a series of modifications, including the replacement of the N-acetyl groups in GlcNAc residues with *N*-sulfo groups by *N*-deacetylase-*N*-sulfotransferases (Ndst), the epimerization of GlcA to IdoA by D-glucuronyl C5-epimerase (Glce), and the addition of sulfate groups at the C2 position of adjacent IdoA residues by 2-*O*-sulfotransferases (Hs2st), the C6 position of GlcNAc residues by 6-*O*-sulfotransferases (Hs6st), and the C3 position of the GlcNS residues by 3-*O*-sulfotransferases (Hs3st) (Figure 3B) [90]. In addition, the mature HS is subjected to remodeling on the cell surface by HS 6-*O*-endosulfatases (Sulf) that selectively remove 6-*O*-sulfo groups from glucosamine residues [90]. Mature HS in the ECM can also be truncated by heparanases secreted by immune cells, which selectively target a variety of trisaccharide sequences and hydrolyze the β-O-linkage between the GlcA and glucosamine residues [91].

Due to substrate specificity and incompletion of the modification by the enzymes, the modifications tend to occur in clusters and generate tremendous structural heterogeneity. The modification patterns form binding sites for many protein ligands, including growth factors, growth factor receptors, and tauopathy-related proteins such as tau, Aβ, and α-syn [92,93,94,95]. In addition, the HS structures are cell-type/tissue/developmental stage-specific, indicating that HS may interact selectively with a fraction of protein ligands to play spatiotemporal regulatory roles under different biological conditions [87,96].

## 6. Heparan Sulfate-Tau Interaction: The Related Structures

HS and heparin, a highly sulfated form of HS, directly bind to tau protein [97,98]. Snow et al. and Su et al. observed by ultrastructural immunolocalization that HS co-localizes with NFTs in neurons in the brains of AD patients [99,100], suggesting that HS interacts with tau in the AD brain. Like other heparin-protein interactions [101,102], the tau/HS interface is dominated by electrostatic interaction mediated through the highly positively charged residues/domain within the tau protein and the highly negatively charged sulfate residues within HS. Further studies determined that the hexapeptides ^275^VQIINK^280^ in R2 and ^306^VQIVYK^311^ in R3 of tau bind to HS [103,104,105,106,107]. In NMR titrations of heparin to K18, L282 displayed the most chemical shift perturbation, suggesting that the two lysines near ^275^VQIINK^280^ in R2, K280 and K281, are likely crucial to heparin and HS binding. A recent study further showed that Arixtra, a sulfated pentamer analog of heparin, bound proline-rich region II (PRR2) with 4 times stronger binding affinity than R2 [108]. These studies indicate that the major HS binding sites within tau protein are in the R2 and PRR2 domains (Figure 4A).

In parallel, several studies examined the HS structural features involved in tau binding. Hasegawa et al. suggested that the overall sulfation level of different GAGs determines their binding affinity to tau, such as heparin having a higher overall sulfation level than HS and showing a higher binding affinity to tau [109]. However, further studies with chemically modified heparins revealed that the binding affinity to tau also depends on heparin’s delicate structure. Removal of *N*- and 6-*O*-sulfation significantly reduced tau-heparin binding, while the removal of 2-*O*-sulfation had less impact [95,106].

Evaluating the interaction of tau with 52 pure and defined HS oligosaccharides on a glycan microarray revealed a striking 3-*O*-sulfation dependence; the binding of tau was increased 1.5- to -5-fold by adding a single 3-*O*-sulfation group on a septamer or dodecamer HS oligosaccharide [93]. Sepulveda-Diaz et al. reported that 3-*O*-sulfated HS interacts with tau and promotes tau phosphorylation [110]. Our recent study also showed that introducing a 3-*O*-sulfate significantly increased the HS binding affinity to tau [93]. These biochemical studies strongly prove that the binding of HS/heparin to tau depends on *N*-, 6-*O*-, and 3-*O*-sulfations.

## 7. The Role of Heparan Sulfate in the Tau-Mediated Pathological Process

HS has been suggested to play an essential role in each stage of the prion-like propagation of tau pathology, including Tau secretion, cell surface binding, internalization, and aggregation [111].

### 7.1. HS in Tau Secretion

Tau is predominately an intracellular protein and has also been found in the extracellular space under both physiological and pathological conditions [112]. Tau is continuously secreted under physiological conditions without cell death, indicating some functional roles of the extracellular tau [113,114,115]. Several studies have demonstrated that tau does not follow the conventional secretory pathway but uses multiple unconventional secretory pathways [116,117]. Merezhko et al. showed that phosphorylated, oligomeric tau clusters on the plasma membrane in neuronal cells and is secreted in the vesicle-free form in an unconventional process, and the secretion was supported by cell surface HSPGs, possibly by facilitating its release after membrane penetration [118]. Katsinelos et al. further delineated that in the cytosol, free tau interacts with phosphatidylinositol 4,5-bisphosphate enriched at the inner leaflet of the plasma membrane, leading to its translocation across the plasma membrane via HSPGs (Figure 4B) [119].

### 7.2. HS in Tau Cell Surface Binding

The association between tau and the cell surface implicates tau protein uptake and related intracellular signaling. Our group observed that tau binds to endothelial cell surface HS. The cell surface binding was inhibited by externally added heparin (Figure 4C), suggesting that HS mediates the cell surface binding of tau, which is consistent with other groups’ findings in which HSPGs mediated the binding of tau to C17.2 cells [111,120]. In addition, knockout of Hs3st1 reduces 3-*O*-sulfation of HS and attenuated endothelial cell surface HS-mediated tau protein binding, supporting the high-affinity tau binding site containing 3-*O*-sulfate (Figure 4D) [93,110].

### 7.3. HS in Tau Internalization

In 2013, Holmes et al. first demonstrated that HSPGs is a critical mediator for tau uptake in mouse neural progenitor cell line [111]. Using differently sized and chemically modified heparins, the same group further determined that tau aggregates required a somewhat specific HS architecture with defined sulfate moieties in the *N*- and 6-*O*-positions, confirmed by reduced tau cellular uptake upon knockout of *Ndst1* or *Hs6st2*, genes responsible for N-sulfation and 6-*O*-sulfation, respectively [95]. Rauch et al. also reported that tau protein internalization depends on 6-*O*-sulfation of HS (Figure 4E) [121].

The 3-*O*-sulfation has been proven to enhance HS-mediated tau internalization. Sepulveda-Diaz et al. reported that 3-*O*-sulfated HS could be internalized into cells where HS interacts with tau and promotes tau phosphorylation [110]. In our recent study, we showed that introducing a 3-*O*-sulfate significantly increased the binding of an HS dodecasaccharide to tau and knockout of HS 3-*O*-sulfotransferase-1 (*Hs3st1*), which reduces about 50% 3-*O*-sulfation in mouse lung endothelial cells [122], significantly decreased tau uptake by the cells (Figure 4D) [93]. Knockout of *Hs3st1* in HEK293T cells appeared not to affect tau uptake [95], which might be due to the low expression of Hs3st1 in this cell line (https://www.proteinatlas.org/ENSG00000002587-HS3ST1/cell+line (accessed on 20 November 2022)). These studies demonstrated that HS facilitates tau protein internalization and further support that the tau-binding HS structure contains *N-*, 6-*O*-, and 3-*O*-sulfations, in agreement with previous biochemical binding studies [95]. In contrast, HSPGs were dispensable for tau protein uptake by primary astrocytes, revealing that HS’s function in facilitating tau internalization is cell-type dependent [95,123].

### 7.4. HS in Tau Aggregation 

HS was found to accumulate with NFTs in the AD brain, suggesting that HS may promote tau aggregation in the brain [100]. Arrasate et al. incubated the isolated PHFs from AD patients with heparinase and found that the PHF morphology was changed after digestion [124]. These observations suggested that HS may facilitate tau protein aggregation to exacerbate tauopathy.

This hypothesis has been supported by the regular in vitro tau aggregation experimental setting, which requires the addition of polyanionic cofactors such as RNA or heparin to initiate the aggregation [125,126,127,128,129,130]. This facilitation depends on the direct binding of heparin to soluble tau monomers, which is thought to neutralize the positive, repulsive charges on tau and allow more contact between monomers [125,126,130]. Townsend et al. examined truncated tau (Δtau187, residue 255–441) aggregation induced with chemically modified heparins [127]. Removal of 6-*O*-sulfation, not 2-*O*-sulfation, reduces heparin’s binding affinity for Δtau187, which in agreement with other study findings, showing 6-*O*-sulfation is required for HS to bind tau protein [95,106]. However, tau aggregation is considerably slower in the presence of 2-*O*-desulfated heparin than with *N*- or 6-*O*-desulfated heparin, indicating that 2-*O*-sulfation contributes more than 6-*O* and *N*-sulfation in facilitating tau aggregation (Figure 4F), apparently due to 2-*O*-sulfation promoting tau primary and secondary nucleation and filament elongation [127]. However, the sulfation pattern dependence has not been examined in the aggregation of full-length tau and phosphorylated tau. It may be different, especially considering the importance of PRR2 in tau binding. In addition, Sepulveda-Diaz, J.E. et al. reported that Hs3st2 increases tau phosphorylation in a cell model and a zebrafish model of tauopathy, showing that HS promotes tau phosphorylation to facilitate tau aggregation indirectly. This is supported by Huynh et al. reporting that Hs3st2 expression induces the cell-autonomous oligomerization of tau, and the phosphorylation of tau, in an in vitro cell model [131].

## 8. Aberrant HSPG Expression in AD and Other Tauopathies

The proteoglycan cores of HSPGs are differentially expressed in AD compared with normal human brains, which has been shown to impact tau internalization [132]. The impact of Alzheimer’s disease on membrane-bound HSPG agrin distribution has been documented since 1999; in normal brains, agrin is in the soluble fraction of detergent extracted samples, while in AD brains, agrin shifts to the insoluble fraction [133]. One study found that genes associated with syndecan-4 (*SDC4*), followed by *SDC3*, *SDC2*, *agrin*, and *serglycin* (the dominant intracellular PG in immune cells) [134] were consistently overexpressed in AD brains compared with controls [135]. SDC3 has the highest affinity for tau monomer, but interestingly, overexpression of *SDC3* and *-4* in K562 cells (lymphoblast cells with low endogenous HS) did not lead to a greater internalization of tau, but increased fibrillation and accumulation of fibrils on the cell surface [132]. These studies emphasize the impact of tauopathies on the specific core proteins of HSPGs.

Several studies have documented altered HS expression in AD patients (Table 1). Su et al. examined 7 AD patients and 4 age-matched controls and observed that the number and intensity of the HS co-staining with PHFs were denser in AD than in control brains [99]. Shimizu et al. examined 25 AD patients with 10 non-demented elderly patient controls. They detected a 9.3-fold HS increase in the hippocampus and a 6.6-fold increase in gyrus frontalis superior in the AD patients. They also observed that HS is most abundantly expressed in the basement membrane of capillary endothelial cells [136]. Other groups’ studies confirmed the abnormal HS expression in the AD brain [92,137].

AD HS showed significantly altered interaction with heparin-binding proteins. Compared to the control brains, GAGs isolated from the AD brains showed decreased binding to growth factors, such as fibroblast growth factor 2 (FGF-2), brain-derived neurotrophic factor and vascular endothelial growth factor (VEGF), and increased binding to tau, heparin-binding EGF-like growth factor and pleiotrophin [92], reflecting GAG structural alteration in AD brain. This has been supported by a recent study showing multiple sulfated disaccharides (ΔUA2S-GlcNS, ΔUA2S-GlcNAc, ΔUA-GlcNAc6S, ΔUA2S-GlcNAc6S) and a tetrasaccharide with rare 3S (ΔUA-GlcNAc6S-GlcA-GlcNS3S6S) were increased in AD [137]. These increased di- and tetrasaccharides are rich in *N*-, 6-*O*-, and 3-*O*-sulfation that bind tau [137] and may increase tau’s propensity for aggregation.

Consistent with these immunostaining and biochemical analyses, the transcripts of several HS-related genes were up-regulated in the AD brain, including *Hs3st2* in Sepulveda-Diaz, J.E. et al. study [110], *Ndst2*, *Hs3st2*, *Hs3st4* and *Glce* in Huynh et al. study, [92], *Hspe* and *Hspe2* in Garcia et al. study [140], and *Extl3*, *Hs6st1*, H*s3st1*, *Hs3st2*, *Hs3st3A1*, *Hs3stB1*, *Hs6st5* and *Hs6st6* in severe AD in Pérez-López et al. study [139], and down-regulated, including HS 6-*O*-endosulfatase-2 (Sulf2) in Roberts et al. study [138] and Sepulveda-Diaz, J.E. et al. study [110]. The Pérez-López et al. study has shown, so far, the most comprehensive HS gene expression profile in AD study, analyzing all HS biosynthesis and remodeling/degradation genes expression in different AD stages and different brain regions [139]. Overall, the results in Pérez-López’s study correlate HS gene expression with AD pathology. The positive or negative correlation depends on the disease’s severity, the area of the brain regions, and the gene function in HS biosynthesis. The most obvious is the upregulation of *Extl3*, *Hs6st1*, and six of the seven members of the *Hs3st* family in severe AD. These findings revealed that the aberrant HS gene expression might generate more tau-binding sites to enhance HS-facilitated tau aggregation, thereby exacerbating the tauopathy [139]. Furthermore, upregulation of HSPE and HSPE2 in AD brains indicates an active role of HS restructuring during disease progression, as the increased expression correlated closely with each concurrent Braak stage of AD [141].

Although most tauopathy studies focus on AD, several studies examined HS expression in other tauopathies. HS co-deposits with NFTs in PiD, Niemann-Pick disease type C, subacute sclerosing panencephalitis, myotonic dystrophy, and motor neuron disease [105,142], although it remains unknown if HS expression is altered in these tauopathies.

## 9. Future Studies from the HSPG Aspect

Much remains to be investigated to understand the pathogenesis of tauopathies better. From the HSPG perspective, most of our knowledge has been gained from AD studies compared to other tauopathies. We have learned that HS critically regulates tau protein secretion, internalization, aggregation, and phosphorylation. We also learned some structural features of the HS motifs that interact with tau protein. However, we are far from clearly defining the roles of HSPGs in all tauopathies and several questions remain, including: 

(1). Cell-type specific roles of HS in tauopathies. The structure of HS is cell-type/developmental/disease-stage dependent, and the biological functions of HSPGs are also location dependent, such as cell surface-anchored vs. in the extracellular matrix. HS on the surfaces of different CNS cell types may also have different roles; e.g., one may hypothesize that HS on neurons may promote the transcellular spread of tau. In contrast, HS on glial cells may promote tau degradation and clearance. It will be essential to understand the spatial and temporal regulatory processes and roles of HS in the pathogenesis of tauopathies, including in tau secretion, internalization, aggregation/deposition, posttranslational modification, and pathological prion-like propagation.

(2). The delicate HS structures that bind tau protein. It has been demonstrated that the tau-binding HS motifs contain *N-,* 6*-O*-, and 3-*O*-sulfation with 2-*O*-sulfation in debate. However, their chemical composition, and more importantly, their delicate modification patterns, are unknown. Successful delineation of the tau-binding site fine structure will open the door to a better understanding of the structure-function relations of HS in interaction with tau proteins, including its six normal isoforms and truncated forms with and without posttranslational modifications. This new information will significantly aid drug design to treat the various tauopathies.

(3). Testing if pharmacological inhibition of HS-tau interaction will ameliorate tauopathy. An early study reported that the treatment with low-molecular-weight heparin prevented abnormal tau protein formation in rat hippocampus [143]. Other studies have observed that heparin, heparin-like molecules (heparinoids and oligosaccharides) competitively inhibit cellular tau uptake in vitro and in vivo [95,111,144] and decrease tau-induced cell toxicity [145]. These findings revealed HS as a promising therapeutic target to inhibit the progression of tauopathies.

Several strategies have been studied targeting HSPGs for therapeutic development, focusing on cancer treatment [146,147]. These include anti-HSPG antibodies, HS antagonists, HS mimetics, and synthetic xylosides. The human monoclonal HS-specific antibody HS20 blocks the activation of the HS-dependent hepatocyte growth factor (HGF)/Met pathway. Consequently, it inhibits HGF-induced hepatocellular carcinoma cell migration, motility, spheroid formation, and liver tumor growth in vivo [148]. Synstatin, a mimetic peptide, inhibits the signaling complex formation between SDC-1, IGF1R, and integrin αvβ3 and attenuates HS-dependent angiogenic VEGF and FGF2 signaling, and blocks tumor angiogenesis in vivo [149,150]. Surfen, bis-2-methyl-4-amino-quinolyl-6-carbamide, had been previously identified as a small molecule antagonist of HS [151]. It neutralizes the anticoagulant activity of unfractionated and low molecular weight heparins and blocks HS-dependent angiogenic FGF2 and VEGF signaling in cultured endothelial cells. Recently, surfen was reported to reduce tumorigenicity of glioblastoma cells in the rat brain [152] and of Ewing sarcoma cells in a zebrafish model [153]. M402 is a rationally engineered, non-cytotoxic HS mimetic and effectively inhibits murine melanoma cell seeding to the lung, a process potentially facilitated by HS, in an experimental metastasis model [154].

Xylosides compete with proteoglycan for HS biosynthetic enzymes and prime GAG chains secreted into the extracellular environment to compete with endogenous proteoglycan-linked GAGs for different binding ligands [155]. By these two mechanisms, xylosides act to block HSPG functions. Xylosides have been shown to inhibit glioblastoma cell viability [156], glioma cell invasion [157], tumor angiogenesis in vitro [158], and various tumor cell line growth in vitro and human bladder carcinoma growth in vivo [159]. These anti-HSPG strategies and available agents give great potential to future tauopathy treatments, exemplified by a recent study from Naini et al. showing that surfen and oxalyl surfen decreased tau hyperphosphorylation and mitigated neuron deficits in vivo in a zebrafish model of tauopathy [160].

In summary, ample evidence supports that HSPGs are critically involved in tauopathy. Heparan sulfate proteoglycans directly bind to co-deposit with tau and modulate tau secretion, internalization, and aggregation. Meanwhile, the expression of heparan sulfate proteoglycans is dysregulated in the disease, and this dysregulation may exacerbate tauopathies. These understandings merge a great potential of targeting heparan sulfate proteoglycan-tau interaction as a novel therapeutic option for the disease.

## Figures and Tables

**Figure 1 biomolecules-12-01792-f001:**
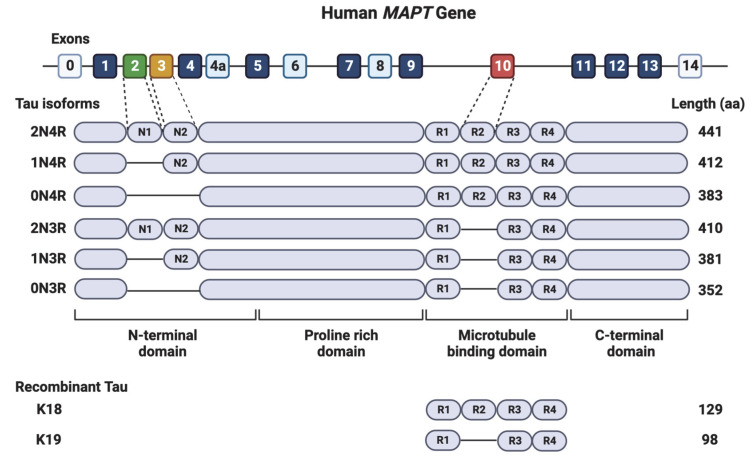
Human *MAPT* gene. The *MAPT* gene encodes human tau and contains 16 exons. E0 and E14 are transcribed but not translated. E1, E4, E5, E7, E9, E11, E12, and E13 are constitutive, and E4a, E6, and E8 are transcribed only in peripheral tissue. The alternative splicing of E2, E3, and E10 generates six tau isoforms seen in normal human brains. The recombinant tau proteins K18 and K19 contain four (R1–R4) and three repeats (R1, R3, R4) of the microtubule-associated domain, respectively.

**Figure 2 biomolecules-12-01792-f002:**
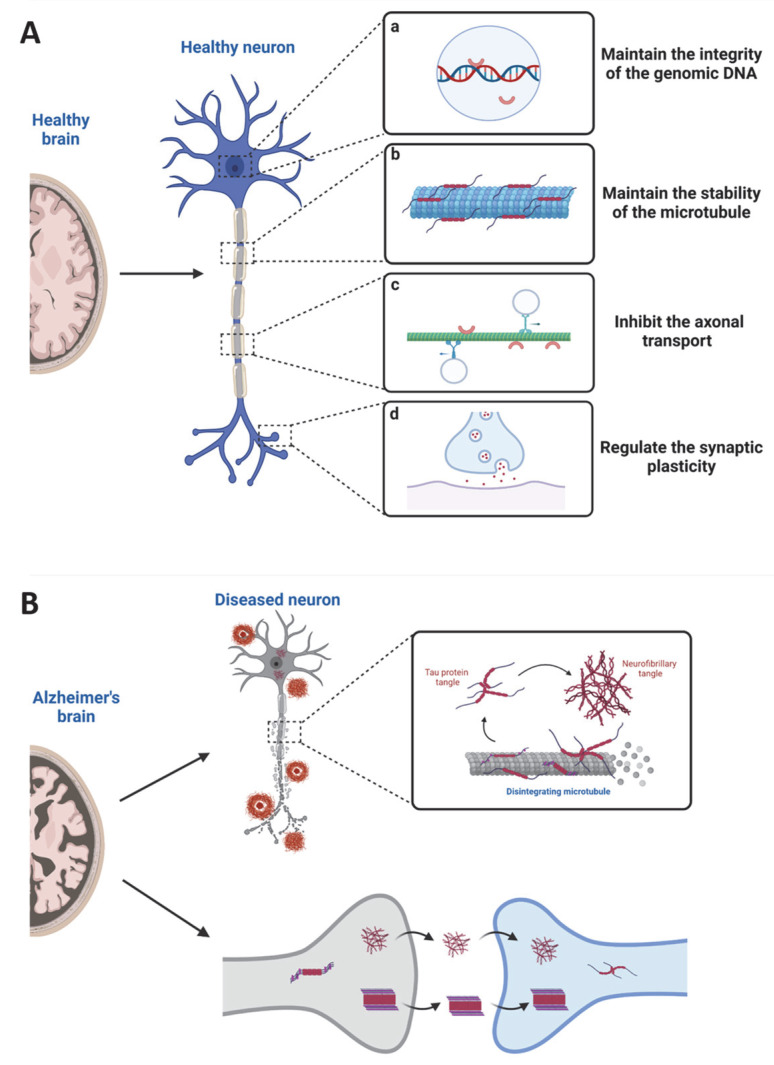
Biological functions of tau in the CNS. (**A**). In the physiological state. Tau protein plays different roles according to its subcellular localization in normal, healthy neurons. In nuclei, tau may function to maintain the integrity of the genomic DNA (a). In the axon, tau maintains the microtubule’s stability and inhibits axonal transport (b and c). In the dendrite, tau functions to regulate synaptic plasticity (d). (**B**). In the pathological state. Under certain stress conditions, the normal tau undergoes hyperphosphorylation and is detached from the microtubule to form tau fibrils and, eventually, the NFTs, leading to neurodegeneration. Meanwhile, NFTs are released from the diseased neurons and uptaken by the neighbor healthy neuron, spreading the disease through prion-like propagation in the CNS.

**Figure 3 biomolecules-12-01792-f003:**
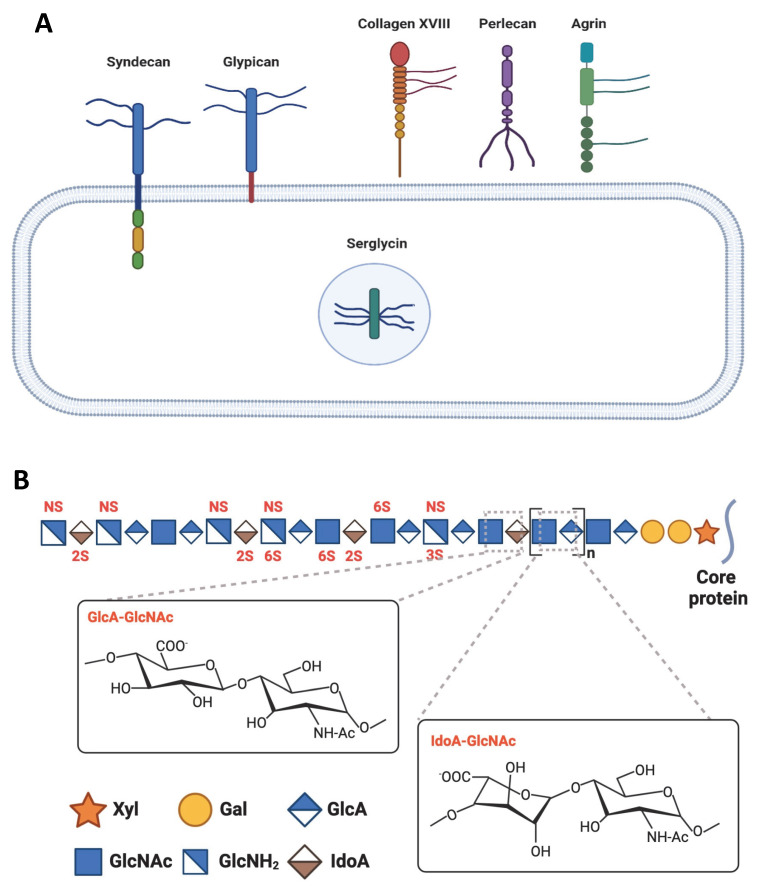
HSPG and HS structure. (**A**). HSPG classification. Mian HSPGs include membrane-bound syndecans and glypicans, extracellular matrix collagen XVIII, perlecan and agrin, and serglycin expressed in the intracellular secretory vesicles. (**B**) HS structure. HS is a linear polysaccharide composed of 20–150 GlcA-GlcNAc/IdoA-GlcNAc repeats with sulfation modifications at NH-Ac, C6, and C3 positions of GlcNAc and C2 position of IdoA. The modifications tend to occur in clusters (sulfated domain). The modification pattern and the sulfated domain arrangement form specific ligand binding sites.

**Figure 4 biomolecules-12-01792-f004:**
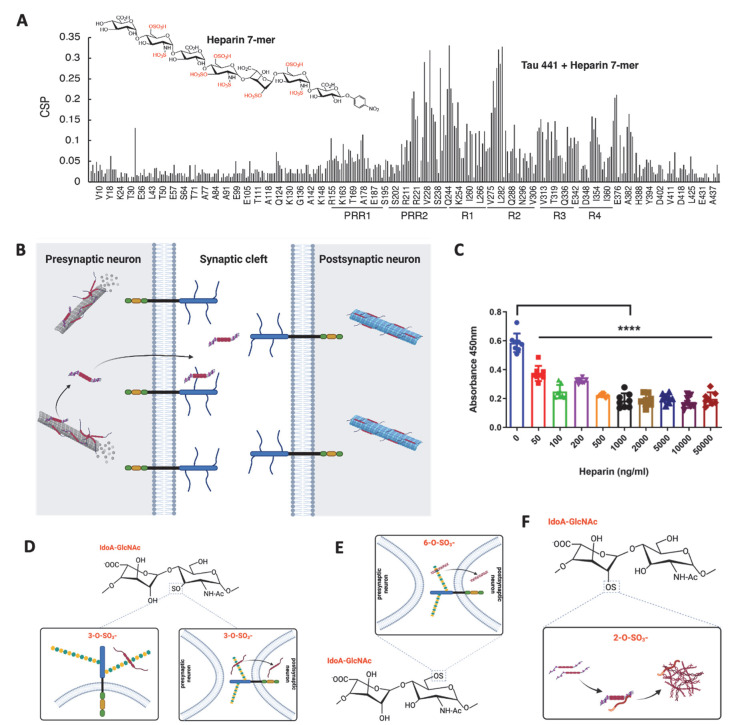
HS-tau interaction. (**A**). The major HS binding sites are located in R2 and PRR2 regions of tau, based on chemical shift perturbation (CS) caused by a heparin 7mer (adapted from Murray et al. [108]). (**B**). Tau secretion from the presynaptic neuron through an unconventional pathway is facilitated by cell surface HSPGs. (**C**). Tau at 50 ng/mL binds to the mouse lung endothelial cell surface. Heparin inhibits tau protein binding to the cell surface dose-dependently and reaches a plateau at 500 ng/mL. ****, *p* value < 0.0001 in ANOVA multiple group analysis (**D**). 3-*O*-sulfation of HS enhances tau protein cell surface binding and cellular uptake. (**E**). 6-*O*-sulfation of HS enhances tau protein cellular uptake. (**F**). 2-*O*-sulfation of HS enhances the tau aggregation.

**Table 1 biomolecules-12-01792-t001:** Altered HS expression and function in AD patients. -, No change; ↑, increase; N/A, not available.

Clinical Diagnosis	Predominant Tau Isoforms	Human Brain Samples	GAGs/Gene Expression in Disease	GAGs Function in Disease	Reference
AD	3R + 4R Tau	7 AD vs. 4 control	HS ↑	N/A	[99]
AD	3R + 4R Tau	N/A	N/A	Helicity of PHFs changed (potential)	[124]
AD	3R + 4R Tau	25 AD vs. 10 control	HS ↑	N/A	[136]
AD	3R + 4R Tau	20 AD vs. 20 control	*Sulf1* -; *Sulf2* ↓	N/A	[138]
AD	3R + 4R Tau	5 AD vs. 5 control	HS ↑; *Ndst2* ↑; *Hs3st2* ↑; *Hs3st4* ↑; *Glce* ↑; *HPSE* ↑	HS-tau binding capacity ↑	[92]
AD	3R + 4R Tau	18 AD vs. 6 control	Altered expression of multiple HS biosynthesis/remodeling genes	N/A	[139]
AD	3R + 4R Tau	5 AD vs. 5 control	HS ↑; 3-o-sulfation ↑	N/A	[137]

## Data Availability

Not applicable.

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
