# Peer review of "Heparan Sulfate Proteoglycans in Tauopathy"

_biomolecules, 2022, doi:10.3390/biom12121792_

Round 1

Reviewer 1 Report

In this review, Zhu and collaborators have done an overview of the role of heparan sulfate proteoglycans in tauopathies. The review is well written, the figures are very clear and well designed and the literature is up-to-date. 

I have only minor comments for the authors that might help to improve their work:

1. The abstract needs to be rewritten in a more precise way. The English is not concise and it is very difficult to follow.

2. Line 32: please consider change Huntington disease for Globular Glial Tauopathy or other disease where tau plays a major role.

2. Line 125: The authors suggest that neurofibrillary tangles can be found in the extracellular space. The authors should provide a reference that clearly support that statement or remove it, as I am not aware of NFTs in the extracellular space.

3. Line 127: change "plaque" to "plaques"

4. Please rephrase line 218, as it is not clear.

5. I would suggest to add a final paragraph to summarize the review and the relevance for the clinical practice.

Author Response

We sincerely appreciate the suggestive comments of Reviewer 1. Please see the following our responses to each of the concerns including additional English editing.

  1. The abstract needs to be rewritten in a more precise way. The English is not concise and it is very difficult to follow.

           - The abstract was rewritten to more precisely reflect the content of the review with a more concise English.

  1. Line 32: please consider change Huntington disease for Globular Glial Tauopathy or other disease where tau plays a major role.

           - Changed as “other disease where tau plays a major role”

  1. Line 125: The authors suggest that neurofibrillary tangles can be found in the extracellular space. The authors should provide a reference that clearly support that statement or remove it, as I am not aware of NFTs in the extracellular space.

           - The critical reference was inserted.

  1. Line 127: change "plaque" to "plaques"

           - The suggested change was made.

  1. Please rephrase line 218, as it is not clear.

           - The sentence was rephrased with addition information from the cited literature to precisely deliver the information.

  1. I would suggest to add a final paragraph to summarize the review and the relevance for the clinical practice.

           -A summary paragraph was added at the end.

Reviewer 2 Report

This is an excellent a topical review on the role of HSPGs in tauopathies. It gives a high quality coverage of the different aspects of the related fields, and perspectives on future research.  This is a valuable and timely addition to the field. Only minor comment is that there are some spelling errors which require careful proof reading. 

Author Response

We sincerely appreciate the suggestive comments of the the reviewer 2. Please see the following our responses with additional English editing.

  1. Only minor comment is that there are some spelling errors which require careful proof reading. 

- We did careful proofreading and found some spelling errors.  These errors were corrected.